# Predictive factors associated with bleeding in atrial fibrillation patients treated with anti-coagulant drugs using a large claims database

Kenji Momo[1,2]*, Kana Shu-toh[1], Makiko Kaneko[3], Nauta Yamanaka[3], Yuji Oto[2], Katsumi Tanaka[2], Masayoshi Koinuma[1], Tadanori Sasaki[2]

1 Faculty of Pharmaceutical Sciences, Teikyo Heisei University, Nakano-ku, Tokyo, Japan, 2 Department of Hospital Pharmaceutics, School of Pharmacy, Showa University, Shinagawa-ku, Tokyo, Japan, 3 JMDC inc., Minato-ku, Tokyo, Japan

* momokenji3@gmail.com

**Data Availability Statement:** All relevant data are within the manuscript and its Supporting Information files.

## Abstract

### Objective

To identify risk factors for bleeding in atrial fibrillation (AF) patients treated with anti-coagulants such as warfarin, apixaban, edoxaban, dabigatran, rivaroxaban using a large claims database.

### Methods

A claims database for 8926 AF patients from 2004 to 2016 was obtained from JMDC. Inc. We performed a retrospective cohort study in 2796 Japanese AF patients with 4-month screening and 12-month observation periods. Polypharmacy was defined as prescription of over six drugs. Logistic regression analysis was conducted after stratification based on the presence and absence of cerebrovascular diseases to detect the predictive factors for bleeding.

### Results

Polypharmacy was observed in 815 of 2796 (29.1%) patients. A total of 371 AF patients (13.3%) experienced bleeding in the 12-month observation period. Bleeding risk assessment using multiple logistic regression analysis revealed that the odds ratio for the number of co-administered drugs in the elderly (age for $\geq$60, $\leq$74) was not significant in those without and with cerebrovascular diseases (1.05 [0.99–1.12], N.S. and 1.10 [0.96–1.27], N.S.). In contrast, in the young (age for <60), the number of co-administered drugs was a significant predictive factor in those without and with cerebrovascular diseases (1.09 [1.03–1.16], p = 0.0054 and 1.20 [1.05–1.36], p = 0.0059). Other observed predictors were "history of bleeding" in young and elderly, but "polypharmacy" and "start from warfarin" were observed in only young.

### Conclusion

We determined the bleeding risk in the clinical setting using a large claims database. Physicians and pharmacists need to monitor patients for the initial bleeding signs, particularly in those with these predictive risk factors.

**Funding:** JMDC inc. provided study resources for large health claims data to this study. JMDC inc. only provided raw data to our study and did not play a role in the study design, data collection and analysis, decision to publish, or preparation of the manuscript.

**Competing interests:** MK and NY are affiliated with JMDC inc. There are no patents, products in development or marketed products to declare. This does not alter our adherence to PLOS ONE policies on sharing data and materials.

## Introduction

Anti-coagulant drugs (warfarin, apixaban, edoxaban dabigatran and rivaroxaban) are used in the treatment of atrial fibrillation (AF). Anti-coagulants prevent thromboses which cause cardiovascular event or strokes in AF patients; however, excessive anticoagulation results in bleeding. Package inserts report major bleeding events in approximately 2% in patients treated with warfarin or Direct Oral Anti-Coagulant (DOAC).

In a sub-analysis of warfarin and rivaroxaban in a phase 3 clinical trial, the proportion of bleeding were higher in the group with 7–9 and over 10 co-administered drugs, than in that at with less than 5 drugs for the first-dose of anti-coagulation [1]. Consistent results were reported in a warfarin vs. apixaban phase 3 study of polypharmacy for the risk for bleeding [2]. This suggests that polypharmacy in AF patients treated with anti-coagulants is likely to increase bleeding events.

In Japan, there were approximately 0.7 million AF patients (0.56% of Japanese population) in 2005 and the prevalence is increasing with the increase in life expectancy [3]. The increasing numbers of AF patients is now considered a major global health care problem. A large-scale cohort study from England showed that the socioeconomic status affected the mortality of ischemic stroke or intracerebral hemorrhage adjusted AF [4]. It is important to identify the predictive factors for anti-coagulant-related bleeding in the working-age population, so as to facilitate prediction of social burdens.

We previously reported that anti-coagulants are involved in drug-drug interactions that increase the incidence of bleeding events after starting anticoagulants [5]. Briefly, pharmacokinetic and pharmacodynamic drug-drug interactions were associated with bleeding events in 3290 AF patients according to a large claims database with 3-month observation periods. Nevertheless, the study did not determine the risk factors for bleeding because of the short observation period. Therefore, in the present study, we aimed to: 1) measure polypharmacy in AF patients treated with anti-coagulants, and 2) identify predictive factors associated with bleeding after starting anti-coagulant treatment using a large claims data with 12-month observation period from Japan.

## Methods

### Data source

The large health insurance claims database was developed by JMDC Co. Ltd., Tokyo, Japan [6]. JMDC collects medical and pharmacy claims from more than 50 occupation-based public health insurance agencies for corporate employees and their family members. As of August 2016, the database included 3600000 recipients aged 0–74 years, representing 2.0% of the Japanese population.

### Case identification and pattern analysis for the number of co-administered drugs

Cases were obtained from 8926 AF patients treated with anti-coagulant drugs (warfarin, apixaban, edoxaban, dabigatran and rivaroxaban) between 2004 and 2016 [6]. Exclusion criteria included lack of data regarding the prescription date, 4-month screening period, and 12-month observation period. A total of 2796 patients were analyzed for the frequency of polypharmacy at the first-dose of anti-coagulant administration, and the predictive factors for bleeding events after starting anticoagulants, using the first dose of anticoagulants. Concomitant use of warfarin or DOACs was counted as one drug. Polypharmacy was defined as the use

of over six drugs for the first-dose of anti-coagulant administration in each patient. Injectable and topical agents were omitted from the counts.

## Identification of cases of bleeding and predictive factors associated with bleeding

Bleeding was identified using the target word "bleeding" or "hemorrhage" in the "Japanese standard disease master" within 12 months after starting anti-coagulants [5]; this code was linked to the ICD-10 code. The standard disease master was developed by "The Committee for Controlled Medical Terminology of Japan Association of Medical Sciences" that was responsible for standardizing disease names, and another committee dedicated for assigning codes to the unique disease names. This is set up in the Social Insurance Medical Fee Payment Fund in conjunction with the Medica Information System Development Center [7]. The Ministry of Health, Labour and Welfare indicated the "Japanese standard disease master" as the standard code for information and use in Japan. This master contains approximately 22000 terms and 2000 modifiers.

## Statistical analysis

Univariate analyses (Students t-test, Mann–Whitney U-test and Chi-squares test) of the groups for "bleeding" and "without bleeding" were performed using the patient demographics of 2796 patients. Multivariate logistic analysis was performed for all co-factors to estimate the risk for bleeding after administration of warfarin or DOAC in 2796 patients. Patients were stratified based on the presence or absence of a history of cerebrovascular diseases (ICD-10 code: I60–69) because of the different base-condition in AF patients. We followed standard methods to estimate sample sizes for multiple logistic regression; at least ten outcomes were needed for each independent variable. In the model with cerebrovascular disease, bleeding was observed in 77 of 297 patients; to avoid overfitting, we selected a maximum of eight factors.

Associations with potential predictors for bleeding were built using simultaneous methods for gender, age, anti-coagulants including warfarin or DOACs, co-administration of anti-platelets (with or without), the number of co-administered drugs, history of bleeding (with or without) and drug-drug interactions (with or without) that are known to have potent drug-drug interactions with warfarin (azole antifungals, macrolides, amiodarone, and anti-cancer agents), apixaban (azole antifungals and macrolides), dabigatran (azole antifungals, verapamil, amiodarone and macrolides) and rivaroxaban (azole antifungals and macrolides). The reasons for adding these factors were as follows: sex: difference in body size between males and females, anticoagulants: warfarin is associated with a higher risk for bleeding as compared to DOACs [8], number of co-administered drugs: co-administration of anti-platelet and drug-drug interactions enhance the bleeding risk for anticoagulants [1, 2], age: the elderly are at a higher risk of bleeding, and past history of bleeding risk: clearly increases bleeding risk. We also added interaction terms for the number of co-administered drugs and patient ages. The reason for adding the interaction terms for the model was that both of these factors were potent factors for bleeding in AF patients; however, the severity of physical conditions were very different between younger and elderly AF patients who received polypharmacy. If an interaction was observed, we stratified the age group as young (age <60) and elderly (age ≥60 and ≤74 year), because the age of 60 years is the time for drastic changes in the working environment including factors such as retirement, position retirement or altered employment status.

Multi-co-linearity was determined using Spearman's rank correlation for r >0.6. The model validity was assessed using "lack of fitness (LOF)", that the model fit when the value over 0.2. Data were expressed as medians with ranges or means ± standard deviations. Data analysis was performed using JMP 14® (SAS Institute Inc.,-Cary,-NC,-US) and Microsoft

Excel add-in software "Excel Stat BellCurve" (SSRI Co. Ltd., Tokyo, Japan). The protocol for this study was approved by the Ethics Committee of Teikyo Heisei University. We did not obtain written or verbal informed consent because of using the anonymous large claims data that fully anonymized before you accessed them. Only limited members working in JMDC inc. had access to the original data [6]. Our co-authors could not access the original data.

## Results

### Polypharmacy in AF patients treated with anti-coagulants

A total of 2796 AF patients with (371; male/female: 281/90, 58.6 ± 9.8 y) and without (2425; male/female: 2,093/332, 55.6 ± 9.7 y) bleeding were assessed for prescription patterns of poly-pharmacy, defined as over six drugs (Table 1). In our dataset, edoxaban was not observed based on our study case identification flow. Polypharmacy was observed in 815 of 2796 (29.1%) patients. In patients with bleeding, prescription patterns for anticoagulants indicated a higher use of warfarin (170/371 vs. 846/2425, p < 0.0001), but lower use of DOACs. The number of patients with co-administration of antiplatelet agents (p = 0.0010), history of bleeding (p < 0.0001), and anticoagulant related drug-drug interactions (p = 0.0018) were higher in the bleeding groups (Table 1). The prescription patterns in the study patients, indicated that the median number of co-administered drugs were five (1–18) and four (1–18) in the with and without bleeding groups, respectively (Table 1). The proportions who received polypharmacy with over six drugs were 27.0% (n = 655) and 43.1% (n = 160) in the without and with bleeding groups, respectively (p < 0.0001) (Fig 1). The most frequent numbers of co-administered drugs were two (n = 455) and four (n = 61) in the without and with bleeding groups, respectively.

### Predictive factors associated with bleeding after administration of anticoagulants

In order to determine the factors associated with bleeding, baseline factors such as back-ground, the number of co-administered anticoagulant drugs, anticoagulants related to drug-

**Table 1. Patients characteristics.**

| | | With bleeding | Without bleeding | *p* |
|---|---|---|---|---|
| n (male/female) | | 371 (281/90) | 2425 (2,093/332) | <0.0001 |
| Age (mean±SD) | | 58.6 ± 9.8 | 55.6 ± 9.7 | <0.0001 |
| Number of drugs | | 5 (1–18) | 4 (1–18) | <0.0001 |
| Anticoagulant agent [%] | | | | |
| | Warfarin | 170 [45.8] | 846 [34.9] | <0.0001 |
| | Apixaban | 41 [11.1] | 307 [12.7] | 0.3821 |
| | Dabigatran | 79 [21.3] | 647 [26.7] | 0.0275 |
| | Rivaroxaban | 81 [21.8] | 625 [25.8] | 0.1038 |
| Number of patients co-administered with antiplatelet agent [%] | | 38 [10.2] | 134 [5.5] | 0.0010 |
| Number of patients with history of bleeding [%] | | 71 [19.1] | 40 [1.6] | <0.0001 |
| Number of patients with anticoagulants relating drug-drug interaction [%] | | 27 [7.3] | 86 [3.5] | 0.0018 |
| Number of patients with disease [%] | | | | |
| | Ischemic heart disease | 137 [36.9] | 524 [21.6] | <0.0001 |
| | Diabetes | 163 [43.9] | 734 [30.3] | <0.0001 |
| | Cerebrovascular disease | 77 [20.8] | 220 [9.1] | <0.0001 |
| | Hypertension | 193 [52.0] | 1128 [46.5] | 0.0481 |

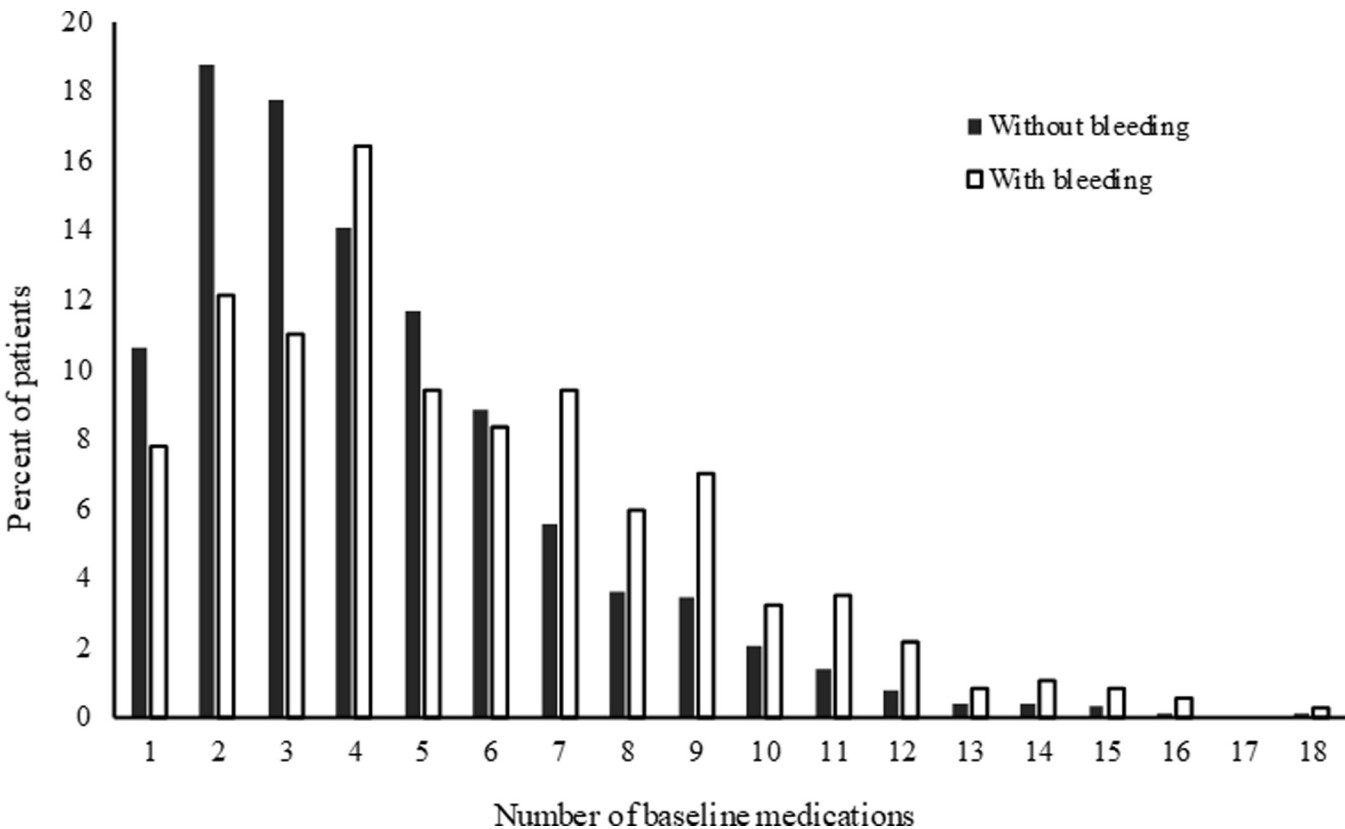

**Fig 1. Frequency of the number of co-administered drugs in 2796 aterial fibrilation patients.**

drug interactions, bleeding history, and the interaction term for the number of co-administered drugs × age were analyzed between with and without bleeding patient groups after stratification without and with a history of cerebrovascular diseases as the first order analysis; we found a difference in base-conditions in AF patients using multiple logistic regression analysis (S1 Table). In the model, female gender, aging for 1 year, number of co-administered drugs, and a history of bleeding were predictive factors for bleeding in the no-cerebrovascular disease group. In the cerebrovascular disease group, the number of co-administered drugs for one co-administered drug, started from warfarin and history of bleeding were predictive factors (S1 Table). In both groups, the interaction term for "number of co-administered drugs × age" was observed significantly (p = 0.0208 and 0.0121, respectively).

Based on data shown in first order analysis, we stratified with and without cerebrovascular diseases and age for as young if <60 years and elderly if aged ≥60 and ≤74 years built in significant predictors for sex, the number of co-administered drugs, anticoagulant agent and past history of bleeding in the first order analysis (Table 2). In the elderly group, the predictive factor for bleeding was a past history of bleeding (without and with cerebrovascular disease: 7.20 [3.40–15.25], p<0.0001 and 11.64 [4.09–33.13], p<0.0001). However, the number of co-administered drugs with DOAC or warfarin were not observed to be a predictive factor for bleeding in our study patients (Table 2). In contrast, in the young group, the number of co-administered drugs was found to be a predictive factor (without and with cerebrovascular disease: 1.09 [1.03–1.16], p = 0.0054 and 1.20 [1.05–1.36], p = 0.0059). In addition, starting anticoagulation with warfarin (without and with cerebrovascular disease: 1.55 [1.06–2.27], p = 0.0042 and 2.98 [1.16–7.66], p = 0.0234) and past history of bleeding (without and with

cerebrovascular disease: 28.83 [12.45–66.76], p<0.0001 and 6.43 [4.31–19.78], p = 0.0006) were also to be significant factors for bleeding (Table 2).

## Discussion

We identified the predictive factors for bleeding after starting anticoagulants in AF patients among young and elderly both in the presence and absence of cerebrovascular disease. In the elderly, only the history of bleeding was a risk factor but in the young, polypharmacy, starting with warfarin, and a history of bleeding were the observed potential predictive factors for bleeding. Anti-coagulant therapy plays an important role in AF patients; the identification of predictive factors associated with bleeding events based on the baseline parameters in AF patients is important for the selection of appropriate anti-coagulants in the real-world clinical settings.

Polypharmacy was observed in 64% of patients in the ROCKT AF study (over five concomitant drugs), 76.5% in the ARISTOLE study (over six concomitant drugs), and 76.9% (over five concomitant drugs) using pharmacy chain dispensing data in Japan [1, 2, 9]. In the present study, 29.1% (over six concomitant drugs) were identified as being administered polypharmacy using claim data (Fig 1). In our study, we used JMDC claim data based on "social insurance" because of the absence of considerable socioeconomic differences as compared to those of other insurance members in Japan. In addition, our subjects, included those aged ≤ 74 years and working in large scale companies, and their family members. The prevalence for AF is high in elderly patients. Therefore, the difference in the prevalence of polypharmacy occurred naturally between the previously mentioned large-scale clinical trials and our study population. Our data demonstrate the reliability of prediction in the under-75 working age population with AF.

**Table 2. Adjusted odds ratio for co-factors associated with bleeding after administration of anti-coagulants using multiple logistic regression analysis stratified based on presence or absence of cerebrovascular disease and age.**

| | | Without cerebrovascular disease (n = 2499) | | | | With cerebrovascular disease (n = 297) | | | |
|---|---|---|---|---|---|---|---|---|---|
| | | With bleeding | Without bleeding | Adjusted Odds ratio (95% CI) | p value | With bleeding | Without bleeding | Adjusted Odds ratio (95% CI) | p value |
| Age ≥ 60, ≤74 | | | | | | | | | |
| Sex, number | Male | 113 | 661 | Reference | | 28 | 91 | Reference | |
| | Female | 41 | 152 | 1.51 (1.01–2.28) | 0.0472 | 13 | 17 | 2.15 (0.84–5.48) | 0.1102 |
| Number of co-administered drugs, median with range, (+1 drug) | | 4 (1–16) | 4 (1–16) | 1.05 (0.99–1.12) | 0.0872 | 6 (1–14) | 5.5 (2–15) | 1.10 (0.96–1.27) | 0.1745 |
| Anticoagulant agent, number | DOAC | 96 | 531 | Reference | | 22 | 73 | Reference | |
| | Warfarin | 58 | 282 | 1.06 (0.73–1.53) | 0.775 | 19 | 35 | 1.99 (0.86–4.60) | 0.1059 |
| Past history of bleeding, number | Without | 137 | 800 | Reference | | 24 | 101 | Reference | |
| | With | 17 | 13 | 7.20 (3.40–15.25) | <0.0001 | 17 | 7 | 11.64 (4.09–33.13) | <0.0001 |
| Age < 60 | | | | | | | | | |
| Sex, number | Male | 109 | 1240 | Reference | | 31 | 101 | Reference | |
| | Female | 31 | 152 | 2.00 (1.25–3.22) | 0.0042 | 5 | 11 | 1.44 (0.41–5.09) | 0.5745 |
| Number of co-administered drugs, median with range, (+1 drug) | | 4.5 (1–18) | 3 (1–18) | 1.09 (1.03–1.16) | 0.0054 | 7.5 (2–15) | 5 (1–15) | 1.20 (1.05–1.36) | 0.0059 |
| Anticoagulant agent, number | DOAC | 72 | 910 | Reference | | 11 | 65 | Reference | |
| | Warfarin | 68 | 482 | 1.55 (1.06–2.27) | 0.0042 | 25 | 47 | 2.98 (1.16–7.66) | 0.0234 |
| Past history of bleeding, number | Without | 115 | 1384 | Reference | | 24 | 100 | Reference | |
| | With | 25 | 8 | 28.83 (12.45–66.76) | <0.0001 | 12 | 12 | 6.43 (4.31–19.78) | 0.0006 |

On stratified detailed analysis, increasing the number of drugs accompanies an increased bleeding risk in young patients with and without history of cerebrovascular diseases (odds ratio: 1.20 and 1.09 for +1 co-administered drugs, respectively p < 0.01) (Table 2); this is similar to findings of previous reports [1, 2]. In contrast, although elderly had higher risk for bleeding as compared to young patients (with and without cerebrovascular diseases: 27.5%/year and 15.9%/year vs. 24.3%/year and 9.1%/year), did not observed increasing the number of drugs accompanies an increased bleeding risk. The possible reasons for this are as follows: 1) elderly patients have poor physical condition, irrespective of the number of drugs used concomitantly and 2) Japanese have a higher bleeding risk than Caucasians, as reported in large-scale phase 3 studies [10].

The advantage of our study was that we identified bleeding risk factors using baseline characteristics of AF patients. Physicians and pharmacists should consider initial bleeding symptoms and counsel patients regarding anti-coagulants accordingly.

We did not assess anti-coagulant medication adherence. Medication adherence is higher in Japan than in other countries [11, 12]. In Japan, low adherence to warfarin was found in approximately 15% [9, 13]. This could partially explain the proportion of bleeding in our study. In the statistical model, LOF was over 0.2 in the model without or with cerebrovascular diseases (S1 Table); however, LOF in the model stratified with cerebrovascular diseases and age were under 0.2 (Table 2). These suggests that the prediction may not have enough in the analysis in Table 2.

## Conclusion

We identified predictive factors associated with bleeding as life-threatening adverse events in patients treated with anti-coagulants. Elderly and younger patients with AF taking several medications should be monitored for initial bleeding events.

## Supporting information

**S1 Table. Adjusted odds ratio for co-factors associated with bleeding after administration of anti-coagulants using multiple logistic regression analysis stratified based on presence or absence of cerebrovascular disease.**
(DOCX)

## Author Contributions

**Conceptualization:** Masayoshi Koinuma.

**Data curation:** Kenji Momo, Kana Shu-toh.

**Formal analysis:** Masayoshi Koinuma.

**Investigation:** Kenji Momo, Kana Shu-toh.

**Methodology:** Kenji Momo.

**Project administration:** Kenji Momo.

**Resources:** Makiko Kaneko, Nauta Yamanaka.

**Supervision:** Tadanori Sasaki.

**Validation:** Masayoshi Koinuma.

**Writing – original draft:** Kenji Momo, Yuji Oto, Katsumi Tanaka.

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
