## [Decision Letter · Decision Letter 0]

18 Mar 2020

PONE-D-20-00210

Predictive factors associated with bleeding in atrial fibrillation patients treated with anti-coagulant drugs using a large claims database

PLOS ONE

Dear Dr. Momo,

Thank you for submitting your manuscript to PLOS ONE. After careful consideration, we feel that it has merit but does not fully meet PLOS ONE’s publication criteria as it currently stands. Therefore, we invite you to submit a revised version of the manuscript that addresses the points raised during the review process.

Your paper was reviewed by three experts in the field. Although the topic is interesting, there are some concerns in the data presentation. Specifically, reviewer 1 is concerned by the statistical analyses. Please read the comments carefully, and address the issues accordingly. I would recommend the authors to consult statistician(s).

We would appreciate receiving your revised manuscript by May 02 2020 11:59PM. To enhance the reproducibility of your results, we recommend that if applicable you deposit your laboratory protocols in protocols.io, where a protocol can be assigned its own identifier (DOI) such that it can be cited independently in the future. For instructions see: http://journals.plos.org/plosone/s/submission-guidelines#loc-laboratory-protocols

We look forward to receiving your revised manuscript.

Kind regards,

Tomohiko Ai, M.D., Ph.D.

Academic Editor

PLOS ONE

Journal Requirements:

2. In ethics statement in the manuscript and in the online submission form, please provide additional information about the patient records used in your retrospective study. Specifically, please ensure that you have discussed whether all data were fully anonymized before you accessed them and/or whether the IRB or ethics committee waived the requirement for informed consent. If patients provided informed written consent to have data from their medical records used in research, please include this information.

3. Thank you for stating the following in the Competing Interests/Financial Disclosure* (delete as necessary) section:

We note that one or more of the authors are employed by a commercial company: name of commercial company.

Reviewers' comments:

Reviewer's Responses to Questions

**Comments to the Author**

1. Is the manuscript technically sound, and do the data support the conclusions?

Reviewer #1: Partly

Reviewer #2: Partly

Reviewer #3: Yes

2. Has the statistical analysis been performed appropriately and rigorously? 

Reviewer #1: No

Reviewer #2: Yes

Reviewer #3: Yes

3. Have the authors made all data underlying the findings in their manuscript fully available?

Reviewer #1: No

Reviewer #2: Yes

Reviewer #3: Yes

4. Is the manuscript presented in an intelligible fashion and written in standard English?

Reviewer #1: No

Reviewer #2: Yes

Reviewer #3: Yes

5. Review Comments to the Author

Reviewer #1: In the manuscript, entitled “Predictive factors associated with bleeding in atrial fibrillation patients treated with anticoagulant drugs using a large claims database,” the authors obtained and analyzed a large claimed database to show the status of polypharmacy including anticoagulants and to identify predictive factors associated with bleeding after starting anti-coagulant treatment. Although their attempt is rather interesting, the conclusion that the interaction term for “number of co-administered drugs * age” is identified as a predictive factor for bleeding would be considered to be inappropriate. Taking it into account that the predictive factor is one of their dominant findings, my recommendation on this manuscript is decided to be rejection. Or at most, major revision.

My concerns are as follows:

The authors concluded that the interaction term for “number of co-administered drugs * age” was identified as a predictive factor for bleeding, the evaluation of which was conducted with logistic regression. The logistic regression model adopted for patients without cerebrovascular disease includes several variables such as age and number of co-administered drugs, along with the interaction term. The p-value of the interaction term (p=0.0208) does represent that the interaction is significant, which indicates that the effect of number of co-administration drugs on bleeding is different at different values of age, but does not represent that the interaction term is a significant predictive factor. The same criticism is applicable for the model for patients with cerebrovascular disease. (The above are based on my recognition. Further review by statistician might be preferable.)

Although the authors conducted detailed analysis to show that the bleeding risk increased with the increase of concomitant drug administration in younger patients, but did not in elderly patients, the data were not shown. Because these results are declared in the ABSTRACT, the data are so important that they should be presented. The data are also important to validate the interaction term and the model itself eventually.

Adequate explanation for the validity of the logistic regression models, especially for the reason for the selection of the variables, is needed, as the authors selected these variables amongst plenty of candidate variables in large claims database.

The entry criteria, referred in Results section, is not specified elsewhere.

Improvements to the English language within the manuscript is desired.

E.g.;

L.27

... after stratification without and with cerebrovascular diseases...

L.145

With bleeding patients, prescription pattern for anticoagulants were high for warfarin, but lower for DOACs.

L.182 Table 2.

Adjusted odds ratio for co-factors associates bleeding after administration of anti-coagulants using multiple logistic regression analysis stratified with and without cerebrovascular disease.

cerebovascular -> cerebrovascular

And so on.

Reviewer #2: #Summary

The authors reported 1) the frequency of polypharmacy in AF patients treated with anti-coagulants; and 2) predictive factors associated with bleeding after starting anti-coagulant treatment using a large claim data in Japan. I think it is an interesting result, but this manuscript does not provide enough data to reach the conclusion. The following issues should be revised:

#Major issues

The authors described as “The interaction term showed that, in young patients, the risk for bleeding was greater as the number of co-administered drugs increased. By contrast, in elderly patients, the risk for bleeding did not increase with increasing number of co-administered drugs (Table 2)” (p.12, L.178-181）. This result is an important finding in this manuscript, but not enough data has been provided to support it. The supporting data should be presented in a table or figure.

#Minor issues

In table 2, odds ratio of “number of co-administration drugs*Age” should be described.

#Other comments

Is the description of “Direct vitamin K inhibitors (DOAC)”（ｐ.4, L.46）correct ?

Is the symbol in “p>0.0001” (p.9, L.146) correct ?

Is the number of “2800” (p.11, L.127) correct ?

References No.8 and No.12 are the same, and their descriptions are duplicated.

Reviewer #3: I read the paper of “Predictive factors associated with bleeding in atrial fibrillation patients treated with anti-coagulant drugs using a large claims database” interestingly. I was particularly interested in discussing the relationship between age and the number of co-administered drugs as predictors of bleeding. I recognized that it was a very useful paper in terms of polypharmacy and medical safety.

Please consider the following points.

Please correct "Shu-toh Kana" to "Kana Shu-toh" in the author part.

How old should elderly patients be considered in this paper? I think it is better to show a specific

age in the discussion.

6. PLOS authors have the option to publish the peer review history of their article (what does this mean?). If published, this will include your full peer review and any attached files.

Reviewer #1: No

Reviewer #2: No

Reviewer #3: No

---

## [Author Response · Author response to Decision Letter 0]

3 Jun 2020

To Reviewer 1

1. The authors concluded that the interaction term for “number of co-administered drugs * age” was identified as a predictive factor for bleeding, the evaluation of which was conducted with logistic regression. The logistic regression model adopted for patients without cerebrovascular disease includes several variables such as age and number of co-administered drugs, along with the interaction term. The p-value of the interaction term (p=0.0208) does represent that the interaction is significant, which indicates that the effect of number of co-administration drugs on bleeding is different at different values of age, but does not represent that the interaction term is a significant predictive factor. The same criticism is applicable for the model for patients with cerebrovascular disease. (The above are based on my recognition. Further review by statistician might be preferable.). Although the authors conducted detailed analysis to show that the bleeding risk increased with the increase of concomitant drug administration in younger patients, but did not in elderly patients, the data were not shown. Because these results are declared in the ABSTRACT, the data are so important that they should be presented. The data are also important to validate the interaction term and the model itself eventually.

We appreciate your pertinent observations and suggestions. A co-author, Dr. Koinuma is a statistician, and he analysed our data. According to the suggestion, we added the analysis in table 3 and added the description (line 33-39, 138-141, 216-229, 244-250).

In general, if interactions are found in categorical variables (eg. Sex), stratified analysis is conducted. In our study, interactions occurred between the variables used, namely, age (continuous variable) and the number of co-administered drugs (continuous variable). Therefore, we presented the results descriptively, as continuous variables usually need to be analyzed as continuous variables. 

However, Kokkinos P et al. (Lancet 2013; 381: 394-99) reported that on changing continuous variables to categorical variables (MET: metabolic equivalents), they found interactions. In accordance with their analysis, we also changed the continuous variables to categorical variables (age cut off value: 60) (Table 3). The age of 60 years was selected based on the time for drastic changes in the working environment such as retirement, position retirement, or employment status (Table 3). In order to avoid over fitting, we added 4 factors (sex, the number of co-administration drugs, anticoagulant agent, and past history of bleeding) to our model in table 3. 

2. Adequate explanation for the validity of the logistic regression models, especially for the reason for the selection of the variables, is needed, as the authors selected these variables amongst plenty of candidate variables in large claims database.

As suggested, we evaluated the model validity using lack of fitness. In the basic model (Table 2), the fit was adequate. On stratified analysis (Table 3), the value did not fit. We have mentioned this in the manuscript at the method and limitation area (line 143-144, 287-291). 

As suggested, the reasons for selecting the included variables have also been mentioned in the manuscript (line 127-132). 

3. The entry criteria, referred in Results section, is not specified elsewhere.

We changed the term “entry criteria” to “case identification flow” to improve clarity (line 159).

3. Improvements to the English language within the manuscript is desired.

E.g.; 

L.27

... after stratification without and with cerebrovascular diseases...

L.145

With bleeding patients, prescription pattern for anticoagulants were high for warfarin, but lower for DOACs.

L.182 Table 2.

Adjusted odds ratio for co-factors associates bleeding after administration of anti-coagulants using multiple logistic regression analysis stratified with and without cerebrovascular disease.

cerebovascular -> cerebrovascular

And so on.

As suggested, we have revised throughout the manuscript, and had it re-edited by a professional English language editing service (Editage). Revised point was wrote in red in the manuscript. 

To Reviewer 2

#Major issues

The authors described as “The interaction term showed that, in young patients, the risk for bleeding was greater as the number of co-administered drugs increased. By contrast, in elderly patients, the risk for bleeding did not increase with increasing number of co-administered drugs (Table 2)” (p.12, L.178-181）. This result is an important finding in this manuscript, but not enough data has been provided to support it. The supporting data should be presented in a table or figure.

We appreciate your pertinent observations and suggestion, and have revised the manuscript accordingly; the stratification in the elderly and young has been included as the second stratified analysis in Table 3 and added the description (line: 33-39, 138-141, 216-229, 244-250).

In general, if interactions are found in categorical variables (eg. Sex), stratified analysis is conducted. In our study, interactions occurred between the variables used, namely, age (continuous variable) and the number of co-administered drugs (continuous variable). Therefore, we presented the results descriptively, as continuous variables usually need to be analyzed as continuous variables. 

However, Kokkinos P et al. (Lancet 2013; 381: 394-99) reported that on changing continuous variables to categorical variables (MET: metabolic equivalents), they found interactions. In accordance with their analysis, we also changed the continuous variables to categorical variables (age cut off value: 60) (Table 3). The age of 60 years was selected based on the timing for drastic changes in the working environment such as retirement, position retirement, or employment status (Table 3). In order to avoid over fitting, we added 4 factors (sex, the number of co-administration drugs, anticoagulant agent, and past history of bleeding) to our model in table 3. 

#Minor issues

In table 2, odds ratio of “number of co-administration drugs*Age” should be described.

The odds ratio for the interaction terms could not be calculated as the changes in the continuous variable * continuous variable values were complex.

As suggested, we added the details of the second stratified analysis for age in the young and elderly to table 3. The odds ratio for the number of co-administered drugs in the elderly (age ≥ 60, ≤74) was 1.05 [0.99 – 1.12], N.S. in the group without cerebrovascular disease and 1.10 [0.96 – 1.27], N.S. in the group with cerebrovascular disease. In contrast, the values in the young were 1.09 [1.03 – 1.16], p=0.0054 in the group without cerebrovascular disease and 1.20 [1.05 – 1.36], p=0.0059 in the group with cerebrovascular disease (Table 3). We add the description (line 33-39, 138-141, 216-229, 244-250).

#Other comments

Is the description of “Direct vitamin K inhibitors (DOAC)”（ｐ.4, L.50）correct ?

As suggested, we have revised this to “Direct Oral Anti-Coagulant” (line 50).

Is the symbol in “p>0.0001” (p.9, L.146) correct ?

We apologize for the error and have revised this from “>” to “<” (line 161).

Is the number of “2800” (p.11, L.127) correct ?

We apologize for the error and have revised this from “2800” to “2796” (line 191).

References No.8 and No.12 are the same, and their descriptions are duplicated.

We apologize for the error and have removed ref. 12 and add ref. 8; the reference numbers have also been changed throughout the manuscript.

To Reviewer 3

Reviewer #3: I read the paper of “Predictive factors associated with bleeding in atrial fibrillation patients treated with anti-coagulant drugs using a large claims database” interestingly. I was particularly interested in discussing the relationship between age and the number of co-administered drugs as predictors of bleeding. I recognized that it was a very useful paper in terms of polypharmacy and medical safety.

Please consider the following points.

Please correct "Shu-toh Kana" to "Kana Shu-toh" in the author part.

We apologize for the error, and have changed "Shu-toh Kana" to "Kana Shu-toh" (line 5).

How old should elderly patients be considered in this paper? I think it is better to show a specific age in the discussion.

As suggested, we defined elderly individuals as those aged ≥ 60. We have added the details of the analysis for stratification in the young and elderly to table 3, and have added the results of the second stratified analysis for age (in the young and elderly) to table 3. The odds ratio for the number of co-administered drugs in the elderly (age ≥ 60, ≤74) was 1.05 [0.99 – 1.12], N.S. in the group without cerebrovascular disease and 1.10 [0.96 – 1.27], N.S. in the group with cerebrovascular disease. In contrast, the values in the young were 1.09 [1.03 – 1.16], p=0.0054 in the group without cerebrovascular disease and 1.20 [1.05– 1.36], p=0.0059 in the group with cerebrovascular disease (table 3). And we added the description (line 33-39, 138-141, 216-229, 244-250).

---

## [Decision Letter · Decision Letter 1]

15 Jul 2020

PONE-D-20-00210R1

Predictive factors associated with bleeding in atrial fibrillation patients treated with anti-coagulant drugs using a large claims database

PLOS ONE

Dear Dr. Momo,

Thank you for submitting your manuscript to PLOS ONE. After careful consideration, we feel that it has merit but does not fully meet PLOS ONE’s publication criteria as it currently stands. Therefore, we invite you to submit a revised version of the manuscript that addresses the points raised during the review process.

Your paper was reviewed by the previous three reviewers. Although your manuscript has been improved, there are still some issues regarding data presentation. Please read the comments by Reviewer 1 carefully, and address the issues accordingly.

We look forward to receiving your revised manuscript.

Kind regards,

Tomohiko Ai, M.D., Ph.D.

Academic Editor

PLOS ONE

Reviewers' comments:

Reviewer's Responses to Questions

**Comments to the Author**

1. If the authors have adequately addressed your comments raised in a previous round of review and you feel that this manuscript is now acceptable for publication, you may indicate that here to bypass the “Comments to the Author” section, enter your conflict of interest statement in the “Confidential to Editor” section, and submit your "Accept" recommendation.

Reviewer #1: (No Response)

Reviewer #2: All comments have been addressed

Reviewer #3: All comments have been addressed

2. Is the manuscript technically sound, and do the data support the conclusions?

Reviewer #1: Partly

Reviewer #2: (No Response)

Reviewer #3: Yes

3. Has the statistical analysis been performed appropriately and rigorously? 

Reviewer #1: Yes

Reviewer #2: (No Response)

Reviewer #3: Yes

4. Have the authors made all data underlying the findings in their manuscript fully available?

Reviewer #1: Yes

Reviewer #2: (No Response)

Reviewer #3: Yes

5. Is the manuscript presented in an intelligible fashion and written in standard English?

Reviewer #1: Yes

Reviewer #2: (No Response)

Reviewer #3: Yes

6. Review Comments to the Author

Reviewer #1: There are still some concerns left, although the authors revised the manuscript almost properly. The concerns are as follows, especially #1-2.

#1-1

The authors added Table 3 and related description according to stratified analysis by the age of 60 years. The revision is to be appropriate, as these results are quite important to support their opinion.

However, this revision generated another concern; Is Table 2 necessary? The authors conducted stratified analysis because interactions are found. Therefore, the predictive factors should be discussed based on the stratified analysis, as authors have done so.

#1-2

No explanation is provided about the predictive factors that I have mentioned. i.e. Is the sentence in ABSTRACT, “Multiple logistic regression analysis revealed the following cerebrovascular disease associated predictors of bleeding in both groups: ...3) interaction term for “number of co-administrered drugs × age”” correct? I don’t think the expression correct. Is regarding an interaction term as a predictor accurate?

#2

Explanations for the validity of their regression models and the reasons for selecting variables are thought to be properly added.

#3

The term “case identification flow” is to be appropriate.

Reviewer #2: (No Response)

Reviewer #3: I read the paper of “Predictive factors associated with bleeding in atrial fibrillation patients treated with anti-coagulant drugs using a large claims database” interestingly. This study aims to assess the polypharmacy of AF patients treated with anticoagulants and use large claims data to identify predictors associated with bleeding after initiation of anticoagulant treatment However, this is an important and meaningful study for medical safety.

It will be clear what the reviewers have pointed out, and readers of the PLOS ONE Journal will be interested.

7. PLOS authors have the option to publish the peer review history of their article (what does this mean?). If published, this will include your full peer review and any attached files.

Reviewer #1: No

Reviewer #2: No

Reviewer #3: No

---

## [Author Response · Author response to Decision Letter 1]

23 Jul 2020

Responses to the Reviewer’s comments (PONE-D-20-00210R1)

We would like to express our gratitude to the reviewers for their insightful comments on our paper. We feel that these comments have helped us to improve the manuscript substantially. Our point-by-point responses to the reviewer’s comments can be found below:

Responses to Reviewer:

Comment #1-1

The authors added Table 3 and related description according to stratified analysis by the age of 60 years. The revision is to be appropriate, as these results are quite important to support their opinion.

However, this revision generated another concern; Is Table 2 necessary? The authors conducted stratified analysis because interactions are found. Therefore, the predictive factors should be discussed based on the stratified analysis, as authors have done so.

Response: 

We appreciate your comment on this point. We removed table 3 and changed it to suppl 

1. According to removing table 3, we revised abstract (p2, l34-50), Method (p7, l 147-

p8, l 149), Result (p12, l 212-232) and Discussion (p18, l265-268, p19, l286-p20, l308, p19, l319-321).

Comment #1-2

No explanation is provided about the predictive factors that I have mentioned. i.e. Is the sentence in ABSTRACT, “Multiple logistic regression analysis revealed the following cerebrovascular disease associated predictors of bleeding in both groups: ...3) interaction term for “number of co-administrered drugs × age”” correct? I don’t think the expression correct. Is regarding an interaction term as a predictor accurate?

Response: 

We appreciate your comment on this point. According to the suggestion for comment 

#1-1, we removed table 3 and mentioned for only the stratified analysis (young and 

elderly) data in revised manuscript. We showed the predictive factor as "history of 

bleeding" in young and elderly, and "polypharmacy" and "start from warfarin" in young 

in revised manuscript (p2, l43-44, p12, l227-232). 

Comment #2

Explanations for the validity of their regression models and the reasons for selecting variables are thought to be properly added.

Response: 

Thank you for your comment.

Comment #3

The term “case identification flow” is to be appropriate.

Response:

Thank you for your comment.

---

## [Decision Letter · Decision Letter 2]

5 Aug 2020

PONE-D-20-00210R2

Predictive factors associated with bleeding in atrial fibrillation patients treated with anti-coagulant drugs using a large claims database

PLOS ONE

Dear Dr. Momo,

Thank you for submitting your manuscript to PLOS ONE. Your paper was evaluated by the previous reviewer. Based upon the comment, your manuscript is improved to meet the reviewer's academic standard. However, some minor changes are suggested, and I would recommend the authors to submit a final version with edits.

We look forward to receiving your revised manuscript.

Kind regards,

Tomohiko Ai, M.D., Ph.D.

Academic Editor

PLOS ONE

Reviewers' comments:

Reviewer's Responses to Questions

**Comments to the Author**

1. If the authors have adequately addressed your comments raised in a previous round of review and you feel that this manuscript is now acceptable for publication, you may indicate that here to bypass the “Comments to the Author” section, enter your conflict of interest statement in the “Confidential to Editor” section, and submit your "Accept" recommendation.

Reviewer #1: (No Response)

2. Is the manuscript technically sound, and do the data support the conclusions?

Reviewer #1: Yes

3. Has the statistical analysis been performed appropriately and rigorously? 

Reviewer #1: Yes

4. Have the authors made all data underlying the findings in their manuscript fully available?

Reviewer #1: Yes

5. Is the manuscript presented in an intelligible fashion and written in standard English?

Reviewer #1: Yes

6. Review Comments to the Author

Reviewer #1: Almost all comments have been addressed properly. However, some sentences are left involving syntax errors.

Abstract

p.2 l.34

Bleeding risk assessment using multiple logistic regression analysis, the odds ratio for the number of co-administered drugs in the elderly (age for ≥60, ≤74) was not significant in those without and with cerebrovascular diseases ...

->

In bleeding risk assessment using multiple logistic regression analysis, …

Or

Bleeding risk assessment using multiple logistic regression analysis revealed that the odds ratio…

Results

p.12 l.227

Data shown in first order analysis, we stratified…

->

Based on data shown in first order analysis, we stratified…

Discussion

p.19 l.267

We identified the predictive factors for bleeding after starting anticoagulants in AF patients among young and elderly and presence and absence of cerebrovascular disease, only the history of bleeding was a risk factor in the elderly, but in the young, polypharmacy, starting with warfarin, and a history of bleeding were the observed potential predictive factors for bleeding.

->

We identified the predictive factors for bleeding after starting anticoagulants in AF patients among young and elderly both in the presence and absence of cerebrovascular disease. In the elderly, only the history of bleeding was a risk factor but in the young, polypharmacy, starting with warfarin, and a history of bleeding were the observed potential predictive factors for bleeding.

7. PLOS authors have the option to publish the peer review history of their article (what does this mean?). If published, this will include your full peer review and any attached files.

Reviewer #1: No

---

## [Author Response · Author response to Decision Letter 2]

10 Aug 2020

Responses to the Reviewer’s comments (PONE-D-20-00210R1)

We would like to express our gratitude to the reviewers for their insightful comments on our paper. We feel that these comments have helped us to improve the manuscript substantially. Our point-by-point responses to the reviewer’s comments can be found below:

Responses to Reviewer:

Thank you for editing English expression. We revised according to the suggestion.

Comment #1-1

Abstract

p.2 l.34

Bleeding risk assessment using multiple logistic regression analysis, the odds ratio for the number of co-administered drugs in the elderly (age for ≥60, ≤74) was not significant in those without and with cerebrovascular diseases ...

->

In bleeding risk assessment using multiple logistic regression analysis, …

Or

Bleeding risk assessment using multiple logistic regression analysis revealed that the odds ratio…

Response: 

According to the suggestion, we revised to “….multiple logistic regression analysis revealed that the odds ratio….”

Comment #1-2

Results

p.12 l.227

Data shown in first order analysis, we stratified…

->

Based on data shown in first order analysis, we stratified…

Response: 

According to the suggestion, we revised to “Based on data ….”

Comment #1-3

Discussion

p.19 l.267

We identified the predictive factors for bleeding after starting anticoagulants in AF patients among young and elderly and presence and absence of cerebrovascular disease, only the history of bleeding was a risk factor in the elderly, but in the young, polypharmacy, starting with warfarin, and a history of bleeding were the observed potential predictive factors for bleeding.

->

We identified the predictive factors for bleeding after starting anticoagulants in AF patients among young and elderly both in the presence and absence of cerebrovascular disease. In the elderly, only the history of bleeding was a risk factor but in the young, polypharmacy, starting with warfarin, and a history of bleeding were the observed potential predictive factors for bleeding.

Response: 

According to the suggestion, we revised to “…among young and elderly both in the presence and absence of cerebrovascular disease. In the elderly, only the history of bleeding was a risk factor but in the young,…”

---

## [Editor Report · Decision Letter 3]

13 Aug 2020

Predictive factors associated with bleeding in atrial fibrillation patients treated with anti-coagulant drugs using a large claims database

PONE-D-20-00210R3

Dear Dr. Momo,

We’re pleased to inform you that your manuscript has been judged scientifically suitable for publication and will be formally accepted for publication once it meets all outstanding technical requirements.

Kind regards,

Tomohiko Ai, M.D., Ph.D.

Academic Editor

PLOS ONE
---

## [Editor Report · Acceptance letter]

19 Aug 2020

PONE-D-20-00210R3 

Predictive factors associated with bleeding in atrial fibrillation patients treated with anti-coagulant drugs using a large claims database 

Dear Dr. Momo:

I'm pleased to inform you that your manuscript has been deemed suitable for publication in PLOS ONE. Congratulations! Your manuscript is now with our production department. 

Kind regards, 

on behalf of

Dr. Tomohiko Ai 

Academic Editor

PLOS ONE